# Atrial Fibrillation Risk and Urate-Lowering Therapy in Patients with Gout: A Cohort Study Using a Clinical Database

**DOI:** 10.3390/biomedicines11010059

**Published:** 2022-12-26

**Authors:** Ching-Han Liu, Shih-Chung Huang, Chun-Hao Yin, Wei-Chun Huang, Jin-Shuen Chen, Yao-Shen Chen, Su-Ting Gan, Shiow-Jyu Tzou, Ching-Tsai Hsu, Hao-Ming Wu, Wen-Hwa Wang

**Affiliations:** 1Division of Cardiology, Department of Medicine, Kaohsiung Armed Forces General Hospital, Kaohsiung 80284, Taiwan; 2Division of Cardiology, Department of Medicine, Tri-Service General Hospital, National Defense Medical Center, Taipei 11490, Taiwan; 3Department of Marine Biotechnology and Resources, National Sun Yat-Sen University, Kaohsiung 80424, Taiwan; 4Institute of Medical Science and Technology, National Sun Yat-Sen University, Kaohsiung 80424, Taiwan; 5Department of Medical Education and Research, Kaohsiung Veterans General Hospital, Kaohsiung 81362, Taiwan; 6Institute of Health Care Management, National Sun Yat-Sen University, Kaohsiung 80424, Taiwan; 7Department of Critical Care Medicine, Kaohsiung Veterans General Hospital, Kaohsiung 81362, Taiwan; 8Department of Administration, Kaohsiung Veterans General Hospital, Kaohsiung 81362, Taiwan; 9Institute of Clinical Medicine, College of Medicine, National Cheng Kung University, Tainan 70101, Taiwan; 10Division of Cardiology, Department of Internal Medicine, Taoyuan Armed Forces General Hospital, Taoyuan 32551, Taiwan; 11Department of Internal Medicine, Division of Cardiology, Kaohsiung Veterans General Hospital, Kaohsiung 81362, Taiwan; 12Health Management Center, Kaohsiung Veterans General Hospital, Kaohsiung 81362, Taiwan; 13Institute of Management, I-Shou University, Kaohsiung 84001, Taiwan; 14Department of Cardiology, Harefield Hospital, Royal Brompton and Harefield NHS Foundation Trust, London UB9 6JH, UK

**Keywords:** allopurinol, febuxostat, gout, incident atrial fibrillation, urate-lowering therapy

## Abstract

Individuals of Asian descent are at higher risk for developing hyperuricemia and gout as compared to Western populations. Urate-lowering therapy (ULT) is an effective treatment for hyperuricemia and gout. It was reported that febuxostat, one of the ULTs, raises the risk of atrial fibrillation (AF) in elderly populations. Nevertheless, this association has not been properly investigated in Asian populations. We aimed to investigate the development of AF after ULT with different drugs in an Asian population. We conducted a retrospective cohort study using the clinical database at Kaohsiung Veterans General Hospital. Patients newly diagnosed with gout between 1 January 2013 and 31 December 2020 and with a documented baseline serum uric acid (sUA) level but no prior diagnosis of AF were identified. Patients were divided into three groups—allopurinol, benzbromarone, and febuxostat users. During the follow-up period, the risks of incident AF following the initiation of ULT with different drugs were assessed. Development of incident AF was noted in 43 (6%) of the 713 eligible patients during the follow-up period (mean, 49.4 ± 26.6 months). Febuxostat-treated patients had a higher prevalence of certain comorbidities (diabetes mellitus, heart failure, and chronic kidney disease) and higher CHA2DS2-VASc scores. Compared with allopurinol, neither febuxostat nor benzbromarone was associated with increased adjusted hazard ratios (HR) for incident AF (HR: 1.20, 95% confidence interval [CI]: 0.43–3.34; HR: 0.68, 95% CI: 0.22–2.08). There was no difference in the risk of incident AF among Asian patients with gout who received febuxostat, allopurinol, or benzbromarone. Further studies are needed to evaluate long-term cardiovascular outcomes in patients receiving different ULT drugs.

## 1. Introduction

Uric acid (UA) is the end-product of purine metabolism in humans, and it plays a significant role in gout and the formation of urolithiasis. Accumulating evidence indicates that UA also plays an etiologic role in various cardiovascular diseases (CVDs), including hypertension, coronary artery disease, heart failure, and atrial fibrillation (AF) [1]. Specifically, hyperuricemia was a significant risk factor for incident AF in both cross-sectional and cohort studies [2,3,4]. The risk of AF is ~60% higher in patients with gout at diagnosis than in age-matched controls, with the potential link to hyperuricemia [5]. Gout is an independent risk factor for incident AF after multivariable adjustment, especially in elderly patients, in some cohort studies [6,7]. Urate-lowering therapy (ULT) is among the mainstays of treatment for patients with hyperuricemia complicated with gout. The American College of Rheumatology Guideline recommends that the serum UA (sUA) level should be lowered to <6.0 mg/dL in patients with gout using a treatment-to-target strategy [8]. However, the beneficial effects of ULT in lowering the risk of cardiovascular (CV) events have not been consistently demonstrated. Furthermore, some randomized clinical trials have raised concerns that febuxostat, a non-purine selective inhibitor of xanthine oxidase, increased the hazard of CV and all-cause mortality [9,10]. This led the Food and Drug Administration (FDA) to issue a public safety alert regarding the first-line use of febuxostat. In elderly patients, febuxostat was associated with an increased risk of incident AF when compared with allopurinol [11]. There have been contradictory results related to the association between febuxostat and increased CV risk in recent large-population studies [12,13,14]. A large prospective study and other real-world data showed that febuxostat was not associated with an increased risk of CV events or all-cause mortality as compared to allopurinol [15,16]. A growing body of evidence no longer supports the increased risk of CVD in febuxostat compared to allopurinol or other ULTs. Some studies conducted in the US and in Korea have reported on an association between racial disparities and adverse effects of the drugs used to treat gout [17,18]. This phenomenon could be attributed to pharmacogenetics regarding different allele frequencies in different populations [19]. Given the high prevalence of gout in the Asian population and the increased risk of developing gout compared with other Western population subgroups, we aimed to analyze the risk of incident AF associated with various ULT drugs in Asians [20,21].

## 2. Methods and Materials

### 2.1. Ethics Statement

We conducted a retrospective cohort study using data recorded in the Kaohsiung Veterans General Hospital clinical database from 2012 to 2021. This study was approved by the Institutional Review Board of Kaohsiung Veterans General Hospital (IRB no. KSVGH22-CT4-12). 

### 2.2. Patient Characteristics

We identified 6665 patients who were newly diagnosed with gout between 1 January 2013 and 31 December 2020 and who had documented baseline sUA levels with no prior diagnosis of AF in Kaohsiung Veterans General Hospital. After the exclusion of 2554 patients who received ULT before the study period, 4111 patients (3391 male, 82%) were enrolled. The exclusion criteria included patients aged ≤40 years and patients who did not receive ULT consistently for more than 2 months. Patients who developed AF after the administration of more than one type of ULT were also excluded (Figure 1). Subsequently, we classified patients into three groups based on the type of ULT they received. The diagnosis of gout was made between 1 January 2013 and 31 December 2020 and was based on the International Classification of Diseases, tenth revision (ICD-10) code M10. xx.

### 2.3. Outcome Analysis

The outcome of interest was the development of incident AF during the follow-up period, which was recognized based on the initial record of the ICD-10 code of I48.xx; this approach has an established median to high diagnostic accuracy of 70–96% [22,23,24]. The recorded follow-up period corresponding to each incident AF episode began on the earliest ULT initiation date during the study period and ended on the earliest date of meeting the incident AF definition or switching to another ULT drug, date of death, or end of the study (31 December 2020), whichever came first. 

In the entire study population, the sUA level was measured within 3 months before the initial diagnosis of gout. After the administration of ULT, which was maintained for at least 2 months, the earliest sUA level was measured for comparison between the three groups. 

### 2.4. Statistical Analysis

Descriptive statistics were used to analyze the baseline demographic data and the distribution of each variable among the study population. Continuous variables are reported as mean ± standard deviation (SD), while categorical variables are reported as proportions. Statistical analyses of continuous variables were performed among the study groups using a one-way analysis of variance (ANOVA), while categorical variables were compared through χ^2^ analysis. The AF-free cumulative survival probabilities were estimated in each group using the Kaplan–Meier method. The log-rank test was used to compare the significance of inequalities with respect to the AF-free survival curves in each group. Multivariate analysis of AF occurrence was performed using the Cox proportional hazards model adjusted by age, body weight, hypertension, congestive heart failure, diabetes mellitus, chronic kidney disease, and prior stroke or transient ischemic attack (TIA). Statistical significance was defined as *p* < 0.05. Hazard ratios (HR) and their 95% confidence intervals (CIs) were used as estimates of relative risk. All statistical analyses were performed with SAS version 9.4 (SAS 9.4, SAS Institute Inc., Cary, NC, USA).

## 3. Results

We identified 713 patients with gout who received ULT with only one of the three analyzed drugs (allopurinol/benzbromarone/febuxostat) consistently for more than 2 months. The majority (591, 83%) of patients were male. As presented in Table 1, the febuxostat group included the oldest patients and the largest proportion of patients with diabetes mellitus compared with the other two groups. There were significant differences in the proportions of patients with congestive heart failure and advanced chronic kidney disease among the three groups.

The follow-up periods were different among the three groups (allopurinol users, 62.2 ± 29.6 months; febuxostat users, 47.2 ± 26.0 months; benzbromarone users, 50.0 ± 25.8 months). The overall mean follow-up duration was 49.4 ± 26.6 months. There was no significant difference in the number of patients who developed incident AF among the three groups (Table 2). Febuxostat or benzbromarone users did not have a higher adjusted likelihood of developing AF compared with allopurinol users (HR: 1.20, 95% CI: 0.43–3.34, *p* = 0.730; HR: 0.68, 95% CI: 0.22–2.08, *p* = 0.492, respectively) (Table 3). Figure 2 presents the AF occurrence-free survival among the three groups. There was no significant difference among the three groups in the incidence of AF (log-rank *p* = 0.502).

The overall mean baseline sUA level was 7.7 ± 2.5 mg/dL, with no significant difference among the three groups (Table 4). The absolute decrease in the sUA level after treatment was larger in febuxostat and benzbromarone users (1.50 and 1.64 mg/dL, respectively) than in allopurinol users (0.65 mg/dL, *p* = 0.002).

## 4. Discussion

We examined the risk of developing incident AF in patients with gout (>40 years old) after ULT with one of the following drugs in a tertiary medical center in Taiwan: allopurinol, febuxostat, or benzbromarone. Our results indicate that compared to the use of allopurinol, the use of febuxostat was not associated with an increased risk of incident AF in patients with gout. To our knowledge, this is the first study to assess the risk of incident AF in Asian patients with gout receiving ULT, while some recent studies from Western countries suggest that the use of febuxostat is harmful in certain populations.

A high level of sUA, the final product of purine metabolism catalyzed by xanthine oxidase, is a strong risk factor for the future development of CVD; this is potentially due to direct pathophysiological effects and a positive correlation with several inflammatory markers [25,26,27,28]. A meta-analysis showed that each 1 mg/mL rise in sUA increased CV mortality by 12% and all-cause mortality by 20% in patients with suspected or definite CAD [27]. During a mean follow-up period of 2–3 years, a longitudinal reduction in sUA levels in gout patients aged ≥ 40 years was associated with a reduced risk of renal function decline but not with that of CVD or diabetes [29]. However, in patients undergoing maintenance hemodialysis, elevated sUA levels were associated with a lower risk of CV mortality [30]. The mechanism underlying this contradictory phenomenon remains elusive.

The association of hyperuricemia and AF was established in the Atherosclerosis Risk in Communities (ARIC) study, which enrolled 15,382 patients, and was confirmed in a nationwide cohort study [2,31]. In a cross-sectional study, the association between hyperuricemia and AF incidence was more pronounced in women than men [4]. Exposure-effect analysis revealed a linear relationship between the sUA level and AF risk in both subjects with normal sUA levels and those with hyperuricemia [32]. However, it remains unclear whether hyperuricemia is only a disease marker or whether it is also a therapeutic target [3]. Gout was found to be associated with an increased risk of incident AF as well as CV morbidity and mortality in the elderly [7]. The link between gout and an increased burden of CVD was multi-factorial, involving higher rates of associated comorbidities (ranging from subclinical atherosclerosis to coronary heart disease) and a higher rate of chronic inflammation [33,34]. 

The use of allopurinol to treat hyperuricemia in the general population or in patients with gout was associated with a modestly reduced risk of all-cause mortality. This relationship was more pronounced in patients with gout than in the general population (19% vs. 11%) [35]. In a retrospective study, allopurinol use, especially at higher doses, was associated with a lower risk of stroke and CV events in patients aged ≥ 65 years with hypertension [36]. However, the association was not consistent, especially in younger populations. In a large population-based cohort study in Asia, the use of allopurinol in patients aged ≥ 40 years with gout was not beneficial in terms of CV protection during a median follow-up period of 5.25 years [37]. On the other hand, the recent Cardiovascular Safety of Febuxostat and Allopurinol in Participants with Gout and Cardiovascular Comorbidities (CARES) trial, conducted at the request of the FDA, raised concerns regarding increased CV mortality in febuxostat users compared to allopurinol users [9]. Specifically, in an earlier 28-week trial involving 1072 patients, there was no significant increase in adverse CV events, including chest pain, coronary artery disease, myocardial infarction, and AF, in patients receiving febuxostat, even at a dose of 240 mg/day, compared to patients receiving allopurinol or placebo [38]. In a large cohort study using Medicare data in the US involving patients aged ≥ 65 years, febuxostat use at the higher dose of 80 mg/day was associated with an increased risk of incident AF in patients ≥ 65 years, as compared with allopurinol < 200 mg/day [11]. The risk was higher during the first 180 days from the initiation of febuxostat treatment but was not observed during longer treatment periods. Another study showed that febuxostat > 80 mg/day or febuxostat > 40 mg/day was associated with a hierarchically increased risk of major CV events or hospitalization, as compared to febuxostat ≦ 40 mg/day. No dose response for CV events was observed in allopurinol [39]. In our study, the average dose of febuxostat users was 54.7 ± 18.0 mg/day, and the average dose of allopurinol users was 120.4 ± 58.7 mg/day. Despite that the average daily dose of febuxostat was not low, no increased risk of incident AF was noted in febuxostat users as compared to allopurinol users in our study. However, caution should be applied when interpreting the data presented by Singh et al. since the study enrolled a mixed population, including those without gout or without a previous history of CVD, in whom the detection of clinical AF might be delayed. The prevalence of heart failure and coronary artery disease—two conditions that increase the frequency of symptomatic AF—was higher in patients receiving febuxostat [40]. This fact might confound the study results to interpret febuxostat as having possible harmful CV effects. In our study, the prevalence of heart failure was different across the three groups. Although a multivariate Cox proportional hazards regression analysis of AF occurrence adjusted by heart failure and other comorbidities was carefully done, the underpinning interactions and pathophysiologic changes driving the co-development of AF and heart failure are hard to evaluate in this study [41]. Furthermore, the level of sUA has a prognostic impact on patients with heart failure across the spectrum of left ventricular ejection fraction. Future studies should address this issue by considering the long-term status of sUA when exploring the ULT effect on CV outcomes [42].

Our study has several strengths worth mentioning. Since an assessment of the protective or harmful effects of ULT with different drugs would not be complete without a sound knowledge of the sUA levels in treated patients, we evaluated laboratory test results, including serial sUA levels as well as the renal function status, before and after the initiation of ULT and confirmed that the risk of incident AF was equivalent among groups with similar pretreatment sUA levels. The three ULT drugs had different urate-lowering capabilities, which was consistent with the observations in our clinical practice. Thus, it can be inferred that the ULT drug should be chosen based on the patient’s refractory hyperuricemia status, age, or underlying comorbidities, including renal function impairment or liver disease. Based on the current evidence, no specific ULT should be defined as the first-line therapy in patients with gout. Second, since the survival curves between febuxostat and allopurinol users began to diverge around 30 months after the initiation of ULT according to the previous study, our study aims to include a relatively young population who received ULT with only one drug [43]. It is our advantage to involve a relatively long observation period (mean follow-up period, 49.4 ± 26.6 months) so that we could monitor the patients with gout for incident AF to potentially occur. In all the patients newly diagnosed with gout (n = 6665), the average time to AF onset was 29.3 ± 23.3 months. The average follow-up times in the three ULT groups were 62.2 ± 29.6 months, 47.2 ±26.0 months, and 50.0 ± 25.8 months, respectively. Therefore, the mean follow-up time in our study seemed adequate for AF occurrence to be noted. Finally, by enrolling patients being treated with only one type of ULT drug before the development of AF and through the application of multivariable adjustments for important AF risk factors, we could evaluate each drug’s effect on AF incidence more specifically.

Our study has several limitations. This was a retrospective cohort study, and the possibility of unknown confounding factors cannot be completely excluded despite our attempt to perform multivariate adjustments for many traditional AF risk factors. Any residual confounding may bias the results of hypothesis testing toward the null hypothesis. For example, some drugs can potentially cause AF, especially in patients with comorbidities [44]. The casualty of drug-induced AF is difficult to ascertain under this condition, which is further complicated by the lack of a biomarker of drug-induced AF. Second, the major limitation of this study was the lack of negative control, i.e., patients with untreated gout, which would have greatly elucidated the inherent risk of AF as a result of gout as a reference for ULT-related AF. Third, the lack of electrocardiographic data might have led to the underdiagnosis of AF, and the use of the ICD-10 code algorithm rendered the differentiation between paroxysmal and non-paroxysmal AF impractical. The potential misclassification bias in the diagnosis of gout cannot be excluded since the identification of gout could sometimes be based on diagnosis by the physician and not meeting the gold standard for diagnosis, which involves the identification of monosodium urate crystals in synovial fluid or tophus aspirates. Moreover, selection bias was unavoidable since this is a retrospective study that relies on hospital records. Finally, we could only define the drug exposure date based on the pharmacy dispensing records, while true medical adherence remains unknown.

## 5. Conclusions

The use of febuxostat in middle-aged to elderly Asian patients (>40 years) with gout was not associated with a higher risk of AF than that of allopurinol or benzbromarone, although these drugs have different urate-lowering capabilities. Therefore, we should not prohibit the use of a more potent drug for ULT, such as febuxostat, to treat gout patients with refractory hyperuricemia.

## Figures and Tables

**Figure 1 biomedicines-11-00059-f001:**
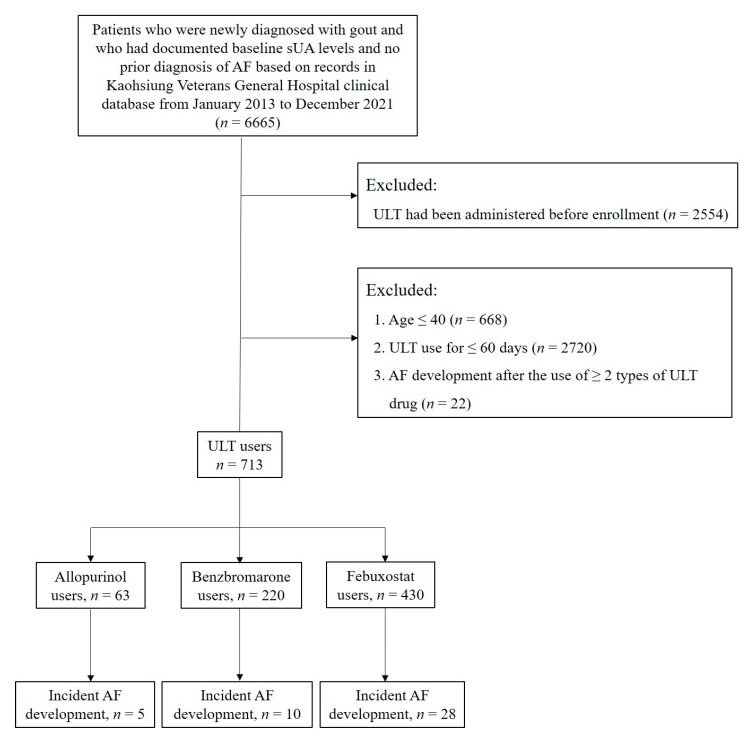
Patient selection flow chart. Abbreviations: AF, atrial fibrillation; sUA, serum uric acid; ULT, urate-lowering therapy.

**Figure 2 biomedicines-11-00059-f002:**
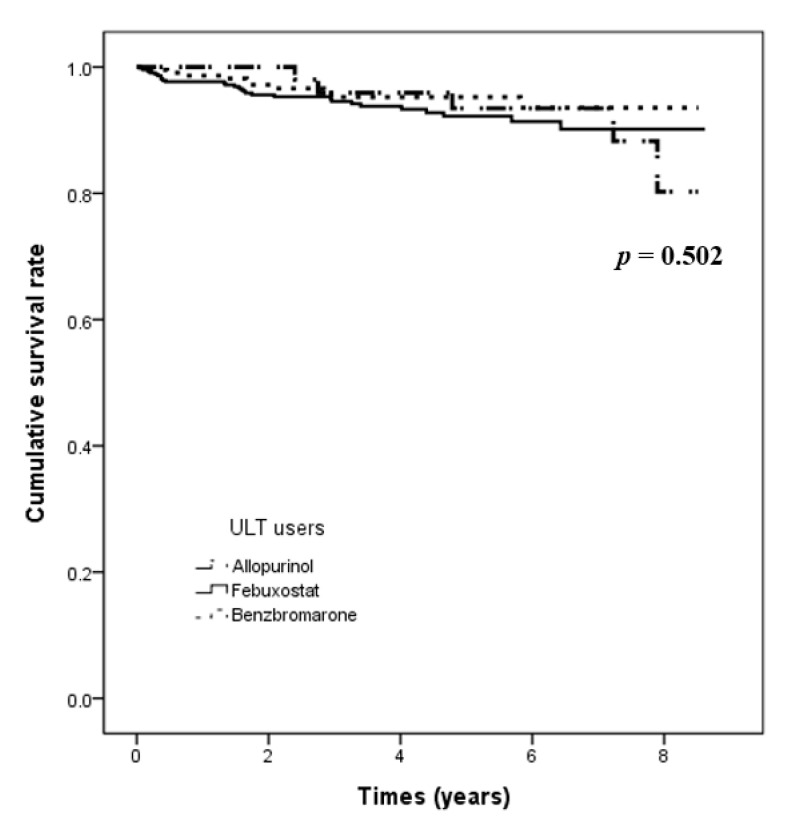
Kaplan–Meier curve for the 9-year AF incidence rate derived through a log-rank test among the three ULT user groups. Abbreviations: AF, atrial fibrillation; ULT, urate-lowering therapy.

**Table 1 biomedicines-11-00059-t001:** Baseline characteristics of patients with gout receiving urate-lowering therapy.

	Total *n* = 713 (%)	Allopurinol *n* = 63 (%)	Febuxostat *n* = 430 (%)	Benzbromarone *n* = 220 (%)	*p*-Value
Age, years	65.9 ± 13.3	64.1 ± 14.1	68.2 ± 13.5	62.0 ± 11.6	<0.001 *
Men	591 (83)	56 (89)	355 (83)	180 (82)	0.405
Drinking	242 (34)	19 (30)	143 (33)	80 (36)	0.586
Smoking	274 (38)	22 (35)	156 (36)	96 (44)	0.158
BMI, kg/m^2^	25.4 ± 4.5	25.9 ± 5.0	25.1 ± 4.6	25.9 ± 4.1	0.103
Hypertension	263 (37)	25 (40)	170 (40)	68 (31)	0.087
Hyperlipidemia	437 (61)	33 (52)	271 (63)	133 (61)	0.257
DM	284 (40)	12 (19)	234 (54)	38 (17)	<0.001 *
CAD	225 (32)	13 (21)	135 (31)	77 (35)	0.096
CHF	123 (17)	11 (18)	86 (20)	26 (12)	0.033 *
Stroke	27 (4)	1 (2)	22 (5)	4 (2)	0.072
Peripheral artery disease	117 (16)	14 (22)	71 (17)	32 (15)	0.348
CHA_2_DS_2_-VASc	2.6 ± 1.7	2.7 ± 1.8	2.9 ± 1.7	2.1 ± 1.6	<0.001 *
Laboratory tests					
^†^eGFR, mL/min	53.9 ± 26.0	64.5 ± 25.1	44.2 ± 25.0	69.8 ± 18.1	<0.001 *
Stage of CKD					<0.001 *
Stage 1 or 2	302 (42)	37 (59)	110 (26)	155 (70)	
Stage 3 or 4	411 (58)	26 (41)	320 (74)	65 (30)	

* *p* < 0.05. † eGFR was calculated using the Cockcroft–Gault equation. Various stages of CKD were defined as follows: stage 1, eGFR = 90 mL/min/1.73 m^2^ or higher; stage 2, eGFR = 60–89 mL/min/1.73 m^2^; stage 3, eGFR = 30–59 mL/min/1.73 m^2^; and stage 4, eGFR = 15–29 mL/min/1.73 m^2^. Categorical data are presented as numbers (%), and continuous data are expressed as mean ± standard deviation. Abbreviations: BMI, body mass index; CAD, coronary artery disease; CHF, congestive heart failure; CKD, chronic kidney disease; DM, diabetes mellitus; eGFR, estimated glomerular filtration rate; TIA, transient ischemic attack.

**Table 2 biomedicines-11-00059-t002:** Crude incidence of AF development among patients receiving different urate-lowering drugs.

	All ULT Users	Allopurinol Users	Febuxostat Users	Benzbromarone Users	
	*n* = 713	*n* = 63	*n* = 430	*n* = 220	*p*-Value
Mean follow-up period (months)	49.4 ± 26.6	62.2 ± 29.6	47.2 ± 26.0	50.0 ± 25.8	<0.001 *
AF development (*n*, %)	43 (6)	5 (8)	28 (7)	10 (5)	0.488

* *p* < 0.05. Categorical data are presented as numbers (%); continuous data are expressed as mean ± standard deviation. Abbreviations: AF, atrial fibrillation; ULT, urate-lowering therapy.

**Table 3 biomedicines-11-00059-t003:** Cox proportional hazards model of AF development between patients with gout receiving treatment with different urate-lowering drugs.

	Crude Analysis		Multivariate Analysis ^†^	
	HR (95% CI) ^∫^	*p*-Value ^‡^	HR (95% CI) ^∫^	*p*-Value ^‡^
Allopurinol users	1		1	
Febuxostat users	1.09 (0.42–2.86)	0.863	1.20 (0.43–3.34)	0.730
Benzbromarone users	0.71 (0.24–2.09)	0.709	0.68 (0.22–2.08)	0.492

† Adjusted for age, body weight, hypertension, congestive heart failure, diabetes mellitus, chronic kidney disease, and prior stroke or TIA. ∫ This table emphasizes HR and its 95% CI, which are more clinically meaningful than the *p*-value. ‡ Bonferroni post hoc multiple comparisons were used. Abbreviations: AF, atrial fibrillation; CI, confidence interval; HR, hazard ratio; ULT, urate-lowering therapy.

**Table 4 biomedicines-11-00059-t004:** Changes in sUA level in patients with gout after treatment with different urate-lowering drugs.

Type of ULT	sUA Levels		
	Baseline	After ULT	Change in sUA
Allopurinol users, mg/dL	7.70	7.20	−0.65
Febuxostat users, mg/dL	7.91	6.45	−1.50
Benzbromarone users, mg/dL	7.34	5.83	−1.64
*p*-Value	0.147	0.020 *	0.002 *

* *p* < 0.05. Values are the number of the variables ± standard deviation. Abbreviations: sUA, serum uric acid; ULT, urate-lowering therapy.

## Data Availability

Available upon request.

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
