# Peer review of "Atrial Fibrillation Risk and Urate-Lowering Therapy in Patients with Gout: A Cohort Study Using a Clinical Database"

_biomedicines, 2022, doi:10.3390/biomedicines11010059_

Round 1
Reviewer 1 Report
Summary
Gout is significantly associated with increased risk for AF. Also, incident AF following the initiation of ULT could occur. Development of incident AF was evaluated in a retrospective study cohort in Taiwan, showing 43 (6%) of the 713 eligible patients during the follow-up period (mean, 49.4 ± 26.6 months) developed AF. Febuxostat-treated patients had a higher prevalence of certain comorbidities and higher CHA2DS2-VASc scores. Compared with allopurinol, neither febuxostat nor benzbromarone was associated with increased adjusted hazard ratios (HR) for incident AF. There was no difference in the risk of incident AF among patients with gout who received febuxostat, allopurinol, or benzbromarone. The study results are consistent with the growing body of evidence that the CVD risk associated with febuxostat is not as high as it was initially thought compared with allopurinol.
Introduction:
1. The authors cited literature demonstrating the relationship between SUA and AF. While treating hyperuricemia is not a standard of care, a focus on the role of gout and AF should be emphasized in the introduction. More literature needs to be cited showing the relationship between gout and AF. https://www.nature.com/articles/srep32220; https://www.ncbi.nlm.nih.gov/pmc/articles/PMC5854102/: https://rmdopen.bmj.com/content/4/2/e000712
2. The authors state that guidelines recommend that the serum UA (sUA) level should be lowered to an optimal target of < 5.0-6.0 mg/dL in patients with gout. There is no evidence that <5.0 mg/dL is effective or recommended by ACR. Please make sure you cite the 2020 ACR guidelines. https://pubmed.ncbi.nlm.nih.gov/32391934/
3. Though the US FDA issued a Black Box Warning on Febuxostat, a growing body of evidence that is no longer supports the increased risk of CVD in Febuxostat compared with allopurinol or other ULT.
4. https://ascpt.onlinelibrary.wiley.com/doi/10.1111/cts.13439
5. https://www.thelancet.com/article/S0140-6736(20)32234-0/fulltext
6. The authors stated that some studies conducted in the US and in Korea have reported on an association between racial disparities and adverse effects of the drugs used to treat gout. Though this is correct, this could be attributed to population differences in pharmacogenetic allele frequencies. This citation could help explain this further. https://www.mdpi.com/2673-9879/2/2/11
7. The author stated their rationale for conducting the study. However, the way it’s formulated is very weak. A stronger rationale would be “given the high prevalence of gout in the Asian population and increased risk for developing gout compared with other population subgroups, the study aimed to analyze the risk of incident AF associated with various ULT drugs in Asian patients. https://bmcrheumatol.biomedcentral.com/articles/10.1186/s41927-021-00239-7 https://www.mdpi.com/2075-4426/11/3/231
Materials and Methods
- Please provide data on other ULT such as probenecid. Were there enough patients taking the medication?
- Did the author evaluate the medications the patients were taking that may induce AF? If not, the authors need to acknowledge that other potential drugs may also cause AF. https://www.ncbi.nlm.nih.gov/pmc/articles/PMC5531271/
- Given the CHA2DS2-VAS scores were statistically significant between the three groups, why not adjust for it on the multivariant analysis?
- Regarding the reduction of SUA, the authors need to acknowledge the dose differences used. Are there any data on the mean dose of the drug in reach respective group? The dose-response relationship could not influence the SUA reduction as well as the risk of potential side effects. Please provide more discussion on the dose-exposure risk.
- A major limitation of this study is the lack of negative control, i.e., patients with untreated gout, which would have greatly elucidated the inherent risk of AF as a result of gout vs ULT-induced gout.
- The limitation section needs to include the potential of case misclassification given the nature of retrospective data. misclassification bias could exist since the identification of gout patients was based on physician diagnosis, rather than according to classification criteria or to the gold standard of urate crystal identification. Also, these studies were retrospective in nature and many were based on hospital records, which may introduce selection bias.
- Some English editing is needed for the study section.
Author Response
Responses to Reviewer #1
Thank you for your detailed comments, which are very helpful to improve the manuscript. Our responses to these comments are below and the relevant passages have been incorporated into the revised manuscript and marked in red font.
- Regarding the comment “The authors cited literature demonstrating the relationship between SUA and AF. While treating hyperuricemia is not a standard of care, a focus on the role of gout and AF should be emphasized in the introduction. More literature needs to be cited showing the relationship between gout and AF.
https://www.nature.com/articles/srep32220; https://www.ncbi.nlm.nih.gov/pmc/articles/PMC5854102/: https://rmdopen.bmj.com/content/4/2/e000712”
Response: Thank you for this very insightful comment. According to your suggestion, we have emphasized the relationship between gout and AF in the revised introduction as follows (page 2, line 54-57) “The risk of AF is ~60% higher in patients with gout at diagnosis than in age-matched controls, with the potential link of hyperuricemia (Kuo CF et al., Rheumatology (Oxford). 2016). Gout is an independent risk factor for incident AF after multivariable adjustment, especially in elderly patients, in some cohort studies (Kuo YJ et al., Sci Rep. 2016; JA Singh et al., RMD Open. 2018),” with the citation of the references you suggested. - Regarding the comment “The authors state that guidelines recommend that the serum UA (sUA) level should be lowered to an optimal target of < 5.0-6.0 mg/dL in patients with gout. There is no evidence that <5.0 mg/dL is effective or recommended by ACR. Please make sure you cite the 2020 ACR guidelines. https://pubmed.ncbi.nlm.nih.gov/32391934”
Response: Thank you very much for your comment. We are sorry that we are not updated with the latest therapeutic guidelines for gout. According to your suggestion, we have corrected the description in the revised introduction as follows (page 2, line 58-60) “The American College of Rheumatology Guideline recommends that the serum UA (sUA) level should be lowered to < 6.0 mg/dL in patients with gout using a treatment-to-target strategy (FitzGerald JD et al., Arthritis Care Res (Hoboken) 2020).”
- Regarding the comment “Though the US FDA issued a Black Box Warning on Febuxostat, a growing body of evidence that is no longer supports the increased risk of CVD in Febuxostat compared with allopurinol or other ULT https://ascpt.onlinelibrary.wiley.com/doi/10.1111/cts.13439
https://www.thelancet.com/article/S0140-6736(20)32234-0/fulltext”
Response: Thank you very much for this insightful comment. According to your suggestion, we have added the descriptions in the revised introduction, with the citation of the references you suggested, as follows (page 2, line 69-72) “A large prospective study and other real-world data showed that febuxostat was not associated with an increased risk of CV events or all-cause mortality as compared to allopurinol (IS Mackenzie et al., Lancet. 2020; S. Sawada et al., Clin Transl Sci. 2022). A growing body of evidence no longer supports the increased risk of CVD in febuxostat compared to allopurinol or other ULTs.” - Regarding the comment “The authors stated that some studies conducted in the US and in Korea have reported on an association between racial disparities and adverse effects of the drugs used to treat gout. Though this is correct, this could be attributed to population differences in pharmacogenetic allele frequencies. This citation could help explain this further. https://www.mdpi.com/2673-9879/2/2/11”
Response: Thank you very much for this helpful suggestion. According to your suggestion, we have added the descriptions in the revised introduction, with the citation of the reference you suggested, as follows (page 2, line 74-75) “This phenomenon could be attributed to pharmacogenetics regarding different allele frequencies in different populations (Alrajeh, K.Y. et al., Future Pharmcol. 2022).” - Regarding the comment “The author stated their rationale for conducting the study. However, the way it’s formulated is very weak. A stronger rationale would be “given the high prevalence of gout in the Asian population and increased risk for developing gout compared with other population subgroups, the study aimed to analyze the risk of incident AF associated with various ULT drugs in Asian patients.
https://bmcrheumatol.biomedcentral.com/articles/10.1186/s41927-021-00239-7
https://www.mdpi.com/2075-4426/11/3/231”
Response: Thank you very much for your insightful comment. According to your suggestions, we have rephrased the rationale of this study, with the citation of the reference you suggested, in the revised introduction as follows (page 2, line 76-78) “Given the high prevalence of gout in the Asian population and the increased risk of developing gout compared with other western population subgroups, we aimed to analyze the risk of incident AF associated with various ULT drugs in Asians (Alghubayshi A et al. BMC Rheumatol. 2022; Butler F et al, J Pers Med. 2021). We have also added the descriptions in the revised abstract to strengthen the rationale of conducting this study as follows (page 1, line 26-27)” Individuals of Asian descent are at higher risk for developing hyperuricemia and gout as compared to the western populations.”
- Regarding the comment “Please provide data on other ULT such as probenecid. Were there enough patients taking the medication?”
Response: Thank you very much for this helpful comment. The ULT “probenecid” has not been available for clinical use till now in Kaohsiung Veterans General Hospital. Therefore, there was no patient who used probenecid during the study period. - Regarding the comment “Did the author evaluate the medications the patients were taking that may induce AF? If not, the authors need to acknowledge that other potential drugs may also cause AF.
https://www.ncbi.nlm.nih.gov/pmc/articles/PMC5531271/”
Response: We appreciated this comment very much. The patient number in our study population who had used drugs that may potentially cause AF is relatively few. No patient ever used prednisolone, doxorubicin, daunorubicin, or epirubicin. There were 11 patients who ever used hydrocortisone, 10 patients who used dexamethasone, and 2 patients who used milrinone, with the average days of use being 6.1±6.0, 5.5±4.0, and 4.5±2.1, respectively. We have addressed this issue in the revised limitations, with the citation of the reference you suggested, as follows (page 8, line 288-290)” For example, some drugs can potentially cause AF, especially in patients with comorbidities (Kaakeh Y et al., Drugs. 2012). The casualty of drug-induced AF is difficult to ascertain under this condition, which is further complicated by the lack of a biomarker of drug-induced AF.”
- Regarding the comment “Given the CHA2DS2-VAS scores were statistically significant between the three groups, why not adjust for it on the multivariant analysis?”
Response: Thank you very much for this insightful comment. We perform a Cox proportional hazards model of AF occurrence adjusted only by patients’ CHA2DS2-VAS scores (univariate analysis). The analysis shows that there was no increased adjusted risk of incident AF in febuxostat users vs. allopurinol users (HR=1.10, 95%CI:39-3.13, p=0.854), or in benzbromarone users vs. allopurinol users (HR=0.63, 95%CI:0.20-1.95, p=0.419). Since we have performed a multivariate analysis with the Cox proportional hazards model adjusted by age, body weight, hypertension, congestive heart failure, diabetes mellitus, chronic kidney disease, and prior stroke or transient ischemic attack, many of which are included in the CHA2DS2-VAS scores, overfitting might occur if we subsequently adjust the data by the CHA2DS2-VAS scores. - Regarding the comment “Regarding the reduction of SUA, the authors need to acknowledge the dose differences used. Are there any data on the mean dose of the drug in reach respective group? The dose-response relationship could not influence the SUA reduction as well as the risk of potential side effects. Please provide more discussion on the dose-exposure risk.”
Response: Thank you very much for your comment. We have added the descriptions of the dose-exposure risk of AF in the revised discussions as follows (page 7, line 240-243; page 7, line 243 - page 8, line 244-246) “ Another study showed that febuxostat > 80 mg/day or febuxostat > 40 mg/day was associated with a hierarchically increased risk of major CV events or hospitalization, as compared to febuxostat ≦ 40 mg/day. No dose-response for CV events was observed in allopurinol (Su CY et al., Mayo Clin Proc. 2019).”;” In our study, the average dose of febuxostat users was 54.7±0 mg/day and the average dose of allopurinol users was 120.4±58.7 mg/day. Despite that the average daily dose of febuxostat was not low, no increased risk of incident AF was noted in febuxostat users as compared to allopurinol users in our study.” - Regarding the comment “A major limitation of this study is the lack of negative control, i.e., patients with untreated gout, which would have greatly elucidated the inherent risk of AF as a result of gout vs ULT-induced gout”
Response: Thank you very much for this comment. In response to your comment, we have addressed this major limitation in the revised limitations as follows (page 8, line 290-293)” Second, the major limitation of this study was the lack of negative control, i.e., patients with untreated gout, which would have greatly elucidated the inherent risk of AF as a result of gout, as a reference for ULT-related AF.” - Regarding the comment “The limitation section needs to include the potential of case misclassification given the nature of retrospective data. misclassification bias could exist since the identification of gout patients was based on physician diagnosis, rather than according to classification criteria or to the gold standard of urate crystal identification. Also, these studies were retrospective in nature and many were based on hospital records, which may introduce selection bias.”
Response: Thank you very much for this helpful comment. According to your suggestion, we have added the descriptions in the revised limitations as follows (page 8, line 295-297; page 9, line 298-300)” The potential misclassification bias in the diagnosis of gout cannot be excluded since the identification of gout could sometimes be based on diagnosis by the physician, and not meeting the gold standard for diagnosis, which involves the identification of monosodium urate crystals in synovial fluid or tophus aspirates. Besides, selection bias was unavoidable since this is a retrospective study that relies on hospital records.” - Regarding the comment “Some English editing is needed for the study section.”
Response: We are very much thankful to you for your deep and thorough review. According to your suggestion, we have sent this manuscript for English editing.
The above descriptions are the responses to your comments and suggestions.
Sincerely yours,
Ching-Han Liu, MD
Wen-Hwa Wang, MD

Reviewer 2 Report
Overall, this is a nice assessment of an interesting topic.
Authors conclude that there was no difference in the risk of incident AF among Asian patients with gout who received febuxostat, allopurinol, or benzbromarone.
This conclusion needs to be strengthened, including formal assessment of possible impact of different length of follow up among the 3 groups.
Also, prevalence of heart failure was significantly different across the 3 groups. It would be important to evaluate whether this also had an impact on results. This reviewer understands that, as they are, there seem to be no differences in A-Fib incidence according to treatment. But they might become evident once heart failure is taken into account, for the combined effect of 2 mechanisms. Please, refer to:
-Characteristics, treatment, and outcomes of newly diagnosed atrial fibrillation patients with heart failure: GARFIELD-AF. ESC Heart Failure 8:1139-1149, 2021
-Serum uric acid and outcomes in patients with chronic heart failure through the whole spectrum of ejection fraction phenotypes: Analysis of the ESC-EORP Heart Failure Long-Term (HF LT) Registry. European Journal of Internal Medicine 89:65-75, 2021
Author Response
Responses to Reviewer #2
Thank you for your detailed comment, which is very helpful to improve the manuscript. Our response to this comment is below.
- Regarding the comment “This conclusion needs to be strengthened, including formal assessment of possible impact of different length of follow up among the 3 groups.”
Response: We are thankful for this precious comment. According to your suggestions, we have added the descriptions in the revised discussion to strengthen the impact of follow-up time as follows (page 8, line 275-279) “In all the patients newly diagnosed with gout (n=6,665), the average time to AF onset was 29.3±23.3 months. The average follow-up times in the three ULT groups were 2±29.6 months, 47.2 ±26.0 months, and 50.0±25.8 months, respectively. Therefore, the man follow-up time in our study seemed adequate for AF occurrence to be noted.” - Regarding the general comment “Prevalence of heart failure was significantly different across the 3 groups. It would be important to evaluate whether this also had an impact on results. This reviewer understands that, as they are, there seem to be no differences in A-Fib incidence according to treatment. But they might become evident once heart failure is taken into account, for the combined effect of 2 mechanisms. Please, refer to:
Characteristics, treatment, and outcomes of newly diagnosed atrial fibrillation patients with heart failure: GARFIELD-AF. ESC Heart Failure 8:1139-1149, 2021
Serum uric acid and outcomes in patients with chronic heart failure through the whole spectrum of ejection fraction phenotypes: Analysis of the ESC-EORP Heart Failure Long-Term (HF LT) Registry. European Journal of Internal Medicine 89:65-75, 2021
Response: We appreciated your comment very much. We can confirm that we have performed a multivariate Cox proportional hazards regression analysis of AF occurrence adjusted by age, body weight, hypertension, congestive heart failure, diabetes mellitus, chronic kidney disease, and prior stroke or transient ischemic attack. Therefore, the significant difference in heart failure prevalence among the three groups should have been addressed. According to your suggestion, we have enlightened this issue in the revised discussion, with the citation of references you suggested, as follows (page 8, line 251-259)” In our study, the prevalence of heart failure was different across the three groups. Although a multivariate Cox proportional hazards regression analysis of AF occurrence adjusted by heart failure and other comorbidities was carefully done, the underpinning interactions and pathophysiologic changes driving the co-development of AF and heart failure are hard to be evaluated in this study (Ambrosio G et al., ESC Heart Fail. 2021). Furthermore, the level of sUA has a prognostic impact on patients with heart failure across the spectrum of left ventricular ejection fraction. Future studies should address this issue by considering the long-term status of sUA when exploring the ULT effect on CV outcomes (Ambrosio G et al., Eur J Intern Med. 2021).
The above descriptions are the responses to your comments and suggestions.
Sincerely yours,
Ching-Han Liu, MD
Wen-Hwa Wang, MD

Round 2
Reviewer 1 Report
Thanks for making all the necessary changes. This is a much-improved manuscript. The authors clearly and precisely addressed all the revisions needed.